# Optimized Cellulase-Hydrolyzed Deoiled Coconut Cake Powder as Wheat Flour Substitute in Cookies

**DOI:** 10.3390/foods11172709

**Published:** 2022-09-05

**Authors:** Tan Phat Vo, Nguyen Hong Nhung Duong, Thuy Han Phan, Thanh Phong Mai, Dinh Quan Nguyen

**Affiliations:** 1Laboratory of Biofuel and Biomass Research, Faculty of Chemical Engineering, Ho Chi Minh University of Technology (HCMUT), 268 Ly Thuong Kiet Street, District 10, Ho Chi Minh City 740500, Vietnam; 2Vietnam National University Ho Chi Minh City, Linh Trung Ward, Thu Duc City, Ho Chi Minh City 71300, Vietnam; 3Biobeau Lab Company, Binh Hung Ward, Binh Chanh District, Ho Chi Minh City 71813, Vietnam

**Keywords:** optimization, fiber-enriched cookies, coconut cake powder, response surface methodology, cellulase

## Abstract

Deoiled coconut cake powder (DCCP) was hydrolyzed to reduce the ratio of insoluble/soluble dietary fiber (RIS) by partially converting insoluble dietary fiber to soluble using Celluclast 1.5 L, a commercial cellulase preparation in citrate buffer medium. Firstly, the influence of citrate buffer amount, enzyme concentration, pH, and retention time on the enzymatic hydrolysis efficiency was investigated. Then, response surface methodology (RSM) was employed to optimize the process in which the insoluble and soluble dietary fiber contents were the responses. The results revealed that 10.3 g buffer/g of materials, 3.7 U/g of the materials, and 60 min of retention time were the optimal conditions for the enzymatic hydrolysis to obtain the insoluble and soluble contents of 68.21%db and 8.18%db, respectively. Finally, DCCP or hydrolyzed DCCP (HDCCP) was partially substituted for wheat flour at different replacement ratios in a cookie recipe at 0, 10, 20, 30, and 40%. The cookies with a 10% replacement ratio of hydrolyzed deoiled coconut cake powders had a lower RIS by more than two folds those of DCCP and had the same sensorial score as the control sample. This study proposed that Celluclast 1.5 L effectively reduced RIS by partially converting insoluble to soluble dietary fiber, improving the soluble dietary fiber content in fiber-enriched cookies.

## 1. Introduction

Cookies are popular desserts worldwide and are well-known for their crunchy texture, rich flavor, and delicious taste. According to the USDA food database, the chemical composition of cookies made from oatmeal and raisins contains 14.3 g of lipids, 34.8 g of total sugar, 29.8 g of starch, and just 3.3 g of total dietary fiber [1]. The database indicates that cookies contain high lipid, sugar, and starch levels, but lack dietary fiber and phytochemical substances. The cookie market value in 2019 was 32.12 billion dollars, and fiber-enriched cookies were projected to reach USD 18.22 billion by 2021 and are anticipated to increase with a growing compound annual growth rate (CAGR) of 7.3% in the next six years from 2022 to 2027 [2,3]. The demands for fiber-enriched cookies keep rising as consumers become more aware of the functionality of fiber.

Dietary fiber can prevent constipation, colon cancer, and cardiovascular diseases, as well as decrease glucose and cholesterol absorption [4]. A variety of fiber sources, such as apple pomace [5], green tea powder [6], or wheat malt [7], have been used to partially or entirely substitute wheat flour in cookies. Dietary fiber can be classified into soluble (SDF) and insoluble (IDF) based on its solubility, which can alter the texture and stabilize food [8]. IDF mainly consists of cellulose, hemicellulose, resistant starch, and chitin, while SDF includes pectins, beta-glucans, galactomannans, mucilages, and hemicelluloses [9]. Both have different actions and influence regular gut activity. The dissolution of SDF can form a viscous gel that increases the transit time through the digestive tract, delays gastric emptying, slows down glucose absorption, and acts as a carbohydrate source for microflora in the large intestine [10]. Meanwhile, IDF is not fermented, thus increasing fecal bulk and excretion of bile acids and decreasing intestinal transit time [9]. Therefore, SDF is considered more beneficial from functional and physiological perspectives than IDF [11]. Daily consumption of at least 5 g of soluble fiber can reduce the presence of metabolic syndrome in type 2 diabetes patients by 54% [12].

Deoiled coconut cake powder (DCCP) is the residue of screw pressing to acquire coconut oil from dehydrated coconut kernels and can be considered a potential source of fiber for cookies. The crude fiber content of DCCP is approximately 16%, with a total dietary fiber (TDF) of 47%. The IDF content is approximately 20 times higher than the SDF content in DCCP [7]. Nonetheless, this ratio is more excessive than the recommended ratio intake from The Dietetic Association of 3:1 for promoting human health [8].

The partial conversion of IDF into SDF in pretreatment can reduce its ratio of insoluble/soluble (RIS) before incorporating it into cookies to boost the health benefits of total dietary fibers (TDF). IDF can be converted into SDF by chemical and enzymatic treatment. Indeed, enzymatic hydrolysis is preferred due to its higher yield, low energy requirements, high specificity, lower equipment necessity, mild process conditions, and environmentally friendly process [13,14]. The enzymatic treatment is accompanied by an increase in the free phenolic concentration, water-soluble antioxidant activity, and phenol compound bioavailability [15]. For instance, the endo β-mannanase treatment decreased the degree of polymerization and produced manno-oligosaccharides from defatted copra meals [9]. Abdessalem Mrabet et al. successfully applied defatted copra meals to reduce RIS from nearly 20 to 2–3 after 30 min of hydrolysis using the commercial enzyme Viscozyme^®^ L [13].

Cellulase is a carbohydrase that can be used for the pretreatment of DCCP by hydrolyzing cellulose, an IDF, the main component in plants [16]. Cellulase has various applications in the food industry, such as enhancing the extraction efficiency of fruit juice and bioactive compound in vegetables and herbs [17]. Aktas-Akyildiz et al. used steam explosion followed by enzymatic hydrolysis with Celluclast 1.5 L at 50 °C and enzyme dosage at 200 nkat xylanase/g of bran for 2 h to increase the SDF content of wheat bran by 52%. As a result, wheat bran’s SDF content and the specific volume of bread increased, while crumb hardness decreased [18]. Celluclast 1.5 L was employed to pretreat *Sargassum horneri* at an enzyme dosage of 1% (*v*/*w*) and 50 °C for 24 h to enhance the extraction yield of bioactive compounds. This research revealed that the polysaccharide and sulfate contents of extracts reached the highest at 65.01 and 12.5%, respectively [14].

Response surface methodology (RSM) enables the accurate evaluation of the interaction effect of the independent parameters on the responses; therefore, RSM with the central composite design (CCD) model was chosen to optimize enzymatic hydrolysis in this research [19]. To the best of our knowledge, there has been no research studying the optimization of the hydrolysis process using Celluclast 1.5 L to reduce RIS by partially converting IDF into SDF in DCCP.

In this study, one-factor experiments were conducted to investigate the effect of technical parameters, including the amount of added citrate buffer, enzyme concentration, pH, and hydrolyzing time on the conversion of IDF to SDF in DCCP to create hydrolyzed deoiled coconut cake powders (HDCCP). Then, the CCD model was employed to optimize the enzymatic hydrolysis process. Finally, HDCCP or DCCP was partially substituted for wheat flour with different ratios to produce fiber-enriched cookies. The qualities of the final baked cookies were evaluated by chemical composition, physical appearance, color, and sensory assessment.

## 2. Materials and Methods

### 2.1. Materials

Copra meal residue provided by Yen manufacturer Ben Tre, Vietnam, is brown; copra meal residue was milled and passed through a 40-mesh sieve to obtain DCCP. The chemical composition of DCCP on dried-weight basis was 4.9% moisture, 5.89% protein, and 0.92% lipid, with RIS of 24.25, 76.82% TDF, 73.81% IDF, and 3.01% SDF.

Materials for cookie preparation: Wheat flour was purchased from Dai Phong company, Ho Chi Minh City, Vietnam. The chemical compounds of wheat flour on the dried base were 13.93% moisture, 10.82% protein, and 1.89% lipid, with RIS of 3.34, 2.52% TDF, 1.94% IDF, and 0.58% SDF. The fresh egg was bought from Ba Huan Company, isomalt was purchased from Vikibomi enterprise, butter originated from Pilot, Australia; Acesulfame K was purchased from Vitasweet, China; salt was purchased from The Southern corporation, vanilla originated from Rayner’s, England; and baking soda was purchased from Alsa, France. Analytical chemicals such as absolute ethanol, acetone, diethyl ether, citric acid monohydrate, sodium citrate, Nessler, and DNS reagent were purchased from Sigma-Aldrich, America. BRENNTAG Vietnam Co., Ltd. provides Termamyl SC (alpha-amylase with the activity of 240 U/g, optimum pH and temperature at 6 and 90 °C, respectively), Alcalase 2.5 L (endopeptidase with activity of 2.5 U/g, and optimal temperature from 30 to 65 °C at pH 7–10), glucose amylase (alpha-glucosidase activity of 270 U, pH 4–5, and temperature at 45–60 °C), and Celluclast 1.5 L (cellulase activity with 700 U/g and optimal temperature at 50 °C).

### 2.2. Experimentation Sections

#### 2.2.1. One-Factor Experiments for Hydrolyzing DCCP by Celluclast 1.5 L

One gram of DCCP was placed into an Erlenmeyer flask, then citrate buffer (0.1 M, pH 6) was added into the DCCP and incubated in water bath (ST 15OSA, Bibby Scientific Limited Stone, Staffordshire, UK) at 50 °C. The varied parameters of the process were the amount of added citrate buffer (5, 7.5, 10, 12.5, 15 g buffer/g of materials), enzyme concentration (0, 3, 5, 10, 15 U/g of materials), pH (5, 6, 7), and hydrolyzing time (0, 15, 30, 60, 90 min). After hydrolysis time, temperature was raised to 90° for 15 min to inactivate Celluclast 1.5 L, and then the wet form of HDCCP was obtained. Next, the wet form of HDCCP was dried at 105 °C in the convection machine (ON-01E, JEIOTECH, Singapore) to acquire HDCCP, having a moisture content of 5%. Subsequently, HDCCP was sieved through 40 mesh. The contents of IDF, SDF, and TDF as well as the RIS of HDCCP were calculated.

#### 2.2.2. Experimental Design for the Optimization of the Enzymatic Hydrolyzing Process

The CCD model was used to optimize the hydrolyzing process of DCCP. The three independent factors and three levels of the CCD are represented in Table 1. The three independent factors at three levels (−1, 0, +1) for 15 experiments were chosen to quantify the response data. The conditional range of three independent factors (the amount of added citrate buffer, enzyme concentration, and time) was selected from the results of one-factor experiments. The enzymatic hydrolysis conditions produced the highest SDF, representing the proper condition (corresponding to 0 level). The boundary conditions were the lower and upper points of proper conditions, corresponding to the level of −1 and 1, respectively. A second-order polynomial model was employed to determine the correlation between response data and independent factors utilizing Equation (1):(1)$$Y=B_{0}+\overset{k}{\underset{i=1}{\sum }}B_{i}X_{i}+\overset{k}{\underset{i=1}{\sum }}B_{\mathrm{ii}}X_{i}^{2}+\overset{k}{\underset{i=1}{\sum }}\overset{k}{\underset{j=1}{\sum }}B_{\mathrm{ij}}X_{i}X_{j}$$

In this equation, B_0_, B_i_, B_ii_, and B_ij_ were the regression coefficient for intercept, linear, quadratic, and interaction terms, respectively. X_i_ and X_j_ represented independent factor values, while k expressed the number of independent factors (k = 3). The three independent factors and their three levels were X_1_, added buffer: 7.5, 10, 12.5 g buffer/g of materials; X_2_, enzyme concentration: 3, 5, 10 U/g of the materials; X_3_, retention time: 15, 30, 60 min. Dependent responses (Y) were IDF (% of dried base, (%db) and SDF (% of dried base, (%db)).

The prediction error (%) between predicted values and experimental values was calculated by the Equation (2):(2)$$Prediction\;error=\frac{\left|the\;mean\;of\;measured\;value-predicted\;values\right|}{the\;mean\;of\;measured\;value}\times 100$$

#### 2.2.3. Cookie Preparation

The cookie recipe consisted of 150 g wheat flour, 70 g isomalt, 46.6 g raw whole egg, 70 g butter, 1.6 g sodium bicarbonate, 0.66 g sodium chloride, 0.18 g acesulfame potassium, 0.6 g vanilla extract, and 13 g water. Wheat flour was partially replaced by DCCP or HDCCP to produce fiber-enriched cookies. The samples for each replacement ratio on the basis of wheat flour were coded as follows: DCCP0 (0% DCCP or HDCCP), DCCP10, DCCP20, DCCP30, and DCCP40 (10, 20, 30, and 40% DCCP, respectively); HDCCP10, HDCCP20, HDCCP30, and HDCCP40 (10, 20, 30, and 40%, HDCCP respectively). DCCP0 was considered as the control sample.

During cookie production, whipping and creaming were conducted by a hand mixer (HR1456, Philips Co., Zhuhai, China). Firstly, the raw whole egg was whipped at 200 rpm for 4 min; sodium chloride and acesulfame potassium were solubilized in water. Such solution and isomalt were simultaneously put into the whipped egg. The mixture was continuously whipped (200 rpm, 4 min) and creamed with butter (200 rpm, 4 min). Sodium bicarbonate and vanilla extract were added, then creaming was further mixed (200 rpm, 1 min). The final cream was blended with wheat flour, wheat flour–DCCP, or wheat flour–HDCCP mixture by a stand mixer (SM8005, Ichiban Ltd., Tokyo, Japan). The acquired dough was rolled to a thickness of 4 mm by hand, and cookies were molded by a round shaper of 35 mm diameter, then put on an aluminum tray. Cookies underwent two baking stages at 175 °C for 10 min and 150 °C for 3 min in a baking oven (GL-1126, Gali Co., Ho Chi Minh City, Vietnam). Finally, baked cookies were cooled to ambient temperature for 45 min before packing in polyethylene bags, and cookie samples were stored at room temperature until further analysis was conducted.

### 2.3. Analytical Methods

#### 2.3.1. Chemical Analysis

AOAC methods were used to investigate the chemical compositions of DCCP, HDCCP, and cookies. The moisture content was determined by AOAC 930.15 by drying the samples to constant weights; the lipid content was measured by AOAC 960.39, and the ash content was quantified by AOAC 942.05 [20]. The protein content was quantified by AOAC 984.13 (Kjeldahl–Nessler method) [21]. IDF and SDF were measured by AOAC 991.42 and 993.19, respectively; the ash and protein contents were also measured to calculate IDF and SDF contents; the sum of IDF and SDF quantified total fiber content [22]. The analyzed results of chemical composition, except moisture content, were expressed as the percent on dry basis (%db). The enzyme activity of Celluclast 1.5 L was performed by the Ghose method using the DNS reagent [23]. Water-holding capacity (WHC) and oil-holding capacity (OHC) were measured by the Yajun Zheng method [24].

#### 2.3.2. Physical Analysis of Cookies

The diameter and thickness of cookies were quantified by the Arti Chauhan method [25], and the spread factor was the ratio of diameter/thickness. The hardness of cookies was determined by three-point breaking methods using Model 5543, Instron, USA, with a speed of 0.5 mm/s [26]. The color machine measured the color of cookies, Model CR-300 (Konica Minolta, Japan), using the CIE Lab system (L representing brightness, a representing redness to greenness, b representing blueness to yellowness, ΔE being the color differences) [25].

#### 2.3.3. Sensory Evaluation of Cookies

The overall acceptability of cookies was evaluated by Ming Huang and Hongshun Yang method [27], using the 9-point hedonic scale (9 = extremely like, 5 = neither like nor dislike, 1 = extremely dislike). Sixty untrained panelists were selected among the students at Ho Chi Minh University of Technology (Ho Chi Minh City, Vietnam). They were checked for allergic reactions with wheat protein and cookie-eating times per year. The cookie samples were placed inside covered cups to avoid moisture loss and flavor interference. All cups were coded with a three-digit random number and were presented at once, including the control sample. Water was provided to clean the palate during the evaluation. The consumers assessed the overall acceptability of cookie samples based on their physical appearance, color, odor, and taste.

### 2.4. Statistical Analysis

The experiments were repeated three times, and the acquired results were shown by means ± sd, standard deviation (*n* = 3). Mean values were considered significant when the multiple range test’s probabilities were less than 0.05. One-way ANOVA (analysis of variance) was carried out using statistical software, Statgraphics Centurion XV (Manugistics Inc., Rockville, MD, USA); graphs were constructed by Origin Pro 2022 (Origin Lab, Northampton, MA, USA); and optimization was performed by Design Expert 13 (Stat-Ease, Inc., Minneapolis, MN, USA)

## 3. Results and Discussion

### 3.1. The Influence of Technical Factors on the Fiber Content of DCCP in the Cellulase Treatment

#### 3.1.1. The Influence of the Added Citrate Buffer on the Fiber Content of DCCP

The influence of citrate buffer addition on the fiber contents of DCCP in the cellulase treatment is represented in Figure 1. The lowest RIS (9.22 ± 0.14) was observed with the medium amount of added citrate buffer of 10 g buffer/g of materials. Increasing citrate buffer addition showed higher SDF content and lower IDF content, indicating that citrate buffer solution facilitates the conversion of IDF to SDF. This improvement in enzymatic hydrolysis can be attributed to the citrate buffer solution, facilitating the cellulase dispersion on the cellulose surface of DCCP and partially preventing lignin from absorbing the cellulase [26,28]. However, a further increase in added citrate buffer caused a shrinkage in the SDF content from 7.48 ± 0.08 to 5.67 ± 0.17%db, while TDF and SDF kept dropping. There might be continuous degradation of SDF into oligosaccharides with a polymerization degree of lower than 10. These oligosaccharides cannot be precipitated in absolute ethanol by the enzymatic gravimetric method, decreasing the SDF content [29]. Therefore, a medium amount of added citrate buffer was suggested to obtain the optimal SDF content and lowest RIS.

#### 3.1.2. The Influence of Enzyme Concentration on the Fiber Content of DCCP

Figure 2 illustrates the influence of enzyme concentration on the fiber content of DCCP in the enzymatic hydrolysis process.

The SDF content of DCCP increased more than two-fold as the enzyme concentration increased from 3 to 5 U/g of the materials. The cellulose hydrolysis rate can positively correlate with the cellulase concentration [30]. However, the further increase in enzyme concentration from 5 to 25 U/g of the materials dropped in SDF content to 2.74 ± 0.05%db, while the RIS increased to 19.6 ± 0.42. This phenomenon can be explained by the concurring process of IDF to SDF and SDF to oligosaccharides at high enzyme concentrations. Therefore, the medium enzyme concentration of 5 U/g of the materials was suitable to acquire the optimal SDF content and low RIS.

#### 3.1.3. The Influence of pH on the Fiber Content of DCCP

The influence of pH on the hydrolysis process of DCCP is evaluated in the pH range of 5–7, as shown in Figure 3.

The SDF content of DCCP increased by 1.48 times when pH rose from 5 to 6. A similar trend was also observed in the case of pH rising from 6 to 7. Asp252 and Asp392 located in the cellulase active site act as the nucleophile and electrophile, respectively; such functional groups catalyzed the conversion of IDF to SDF [31]. The pH of the solution affects the nucleophilicity and electrophilicity of Asp252 and Asp392, possibly due to alteration in the proton donation and acceptance. There is a decrease in the proton donation of Asp252 and an increase in the proton acceptance of Asp392 at pH 5, while the reverse trend is observed for pH 7 [30,31]. This phenomenon can lead to a drop in cellulase activity when the pH of the solution is at 5 and 7, respectively. However, the SDF content at pH 5 was 1.3 times as high as that of pH 7, probably due to the hydrolysis of polysaccharides in the weak acidic environment [32]. Therefore, pH 6 was suggested to obtain the optimal SDF content and the lowest RIS.

#### 3.1.4. The Influence of Hydrolyzing Time on the Fiber Content of DCCP

The effect of processing time on fiber content during cellulase treatment is shown in Figure 4. As time increased from 15 to 60 min, the IDF dropped, while SDF increased and TDF did not change. This trend indicated that IDF was converted to SDF, and there was almost no further depolymerization of SDF to smaller units. The RIS values were also the smallest during this time. However, with a further rise in retention time over 60 min, both IDF and SDF contents experienced a decline to 4.13 ± 0.24%db for the retention time of 180 min, leading to reduced TDF and higher RIS. The results reveal that SDF can be further converted to oligosaccharides during hydrolysis processes for a longer time, escaping the precipitation of absolute ethanol [29]. Therefore, 30–60 min retention time was suitable for obtaining the optimal SDF content and lowest RIS.

### 3.2. Optimizing the Cellulase Hydrolysis Process of DCCP

#### 3.2.1. The Regression Models of IDF and SDF

The values of independent factors were selected from the practical means of the cellulase hydrolysis process in Section 3.1.1, Section 3.1.2 and Section 3.1.4, including middle (proper conditions), low, and high (coded 0, 1, and −1, respectively). The pH was fixed due to its optimum for cellulase activity, structure, proton donation, and acceptance. By contrast, the other conditions, such as the amount of added citrate buffer, enzyme concentration, and time, were selected as factors affecting the diffusion and reaction rate of the cellulase. The outcomes obtained from the cellulase hydrolysis process of DCCP are expressed in Table 2, and Table 3 shows the regression coefficients.

The regression coefficients, having a lower *p*-value than 0.05, were employed for constructing regression models (Equations (3) and (4)) for the prediction of response values. The predicted models expressed the F-values of 29.27 and 44.30 for IDF and SDF, respectively; the high values of R^2^ (0.9742, 0.9819) revealed the proper models.


Y_IDF_ = 69.34 − 0.68X_1_ + 0.31X_2_ − 0.62X_3_ + 0.15X_1_X _3_ + 2.31X_1_^2^ + 0.67X_2_^2^
(3)



Y_SDF_ = 7.17 + 0.27X_1_ + 0.49X_3_ − 2.21X_1_^2^ − 1.18X_2_^2^ + 0.57X_3_^2^
(4)


Three-dimensional response surface plots (Figure 5) demonstrate the relation between independent factors and dependent responses; these graphs remain unchanged at the middle level, while the rest of the factors are varied. The second-order polynomial Equations (2) and (3) demonstrated the relation of IDF and SDF, respectively, with independent factors. The IDF content was considerably (*p* < 0.01) affected by X_12_, X_1_, and X_3_ (Table 3), while the interactive effect of citrate buffer addition and time significantly impacted (*p* < 0.05). Three-dimensional response surface plots are built to imagine the effects of three independent factors on the IDF content of DCCP (Figure 5(A1–A3)). It could be illustrated that time and citrate buffer addition positively affected the IDF contents through the enzymatic hydrolysis process.

The SDF content was substantially (*p* < 0.01) impacted by X_12_, X_22_, and X_3_ (Table 3). Three-dimensional response surface graphics are constructed to visualize the influence of three independent factors on the SDF content of DCCP (Figure 5(B1–B3)). It could be shown that the SDF content reached the highest point at 8.18%db, followed by a moderate decrease. Therefore, the optimal conditions of the cellulase process were as follows: citrate buffer addition—10.3 g buffer/g of materials; enzyme concentration—3.7 U/g of the materials; retention time of 60 min, to attain 68.21%db of IDF, 8.18%db of SDF, and 8.33 of RIS (data not shown).

#### 3.2.2. Model Validation

Table 4 depicts the values of practical experiments at optimal conditions in the cellulase process. The reliability of CCD models was verified by conducting experiments at the optimal conditions of cellulase treatment that were picked from 3D surface graphics and regression analysis.

Citrate buffer addition: 10.3 g buffer/g of materials, 3.7 U/g of the materials, and 60 min of retention time were optimal for the enzymatic hydrolysis process to reduce RIS. The forecast values of IDF and SDF were 68.21%db, 8.18%db, and 8.33 of RIS (data not shown), respectively; it could be shown that the forecast values well-matched the values of experiments with low prediction errors (<6%) [33].

### 3.3. The Influence of Partial Replacement on Cookie Properties

#### 3.3.1. Chemical Compounds of HDCCP

The chemical analysis of HDCCP was conducted to examine the variation of chemical composition compared to DCCP. The process with Celluclast 1.5 L raised the SDF content of DCCP by 2.7 times (from 3.01 to 8.18%db) due to cellulosic hydrolysis. The RIS was decreased by 2.9 times, while the protein and lipid content of DCCP did not vary after the cellulase process (data not shown). In contrast, the WHC of DCCP (5.98 ± 0.04 g water/g db) was significantly higher than that of HDCCP (5.48 ± 0.08), while the reverse trend was actually for the case of OHC from 2.21 ± 0.15 to 3.26 ± 0.14 g oil/g db. The hydrolysis process occurs in citrate buffer, being able to partially support the esterification between citrate ions and the hydroxymethyl groups of DCCP polysaccharides, leading to rising the hydrophobicity of obtained HDCCP [34,35].

#### 3.3.2. The Influence of the Partial Replacement of Wheat Flour on the Chemical Composition of Cookie

The influence of incorporating DCCP or HDCCP at different ratios into the cookie formula on the chemical composition of cookies is illustrated in Table 5.

The replacement of DCCP and HDCCP increased ash, IDF and SDF contents, and RIS, while the moisture and lipid contents remained unchanged. The TDF content of DCCP40 and HDCCP40 samples (Table 5) was 6.1 and 5.8 times higher than that of DCCP0. All DCCP- or HDCCP-incorporated cookies expressed a TDF content above 6%db; thus, they could be taken into account as high-fiber food by Codex Alimentarius [36]. The SDF content of HDCCP-incorporated cookies was 1.2 times higher, while RIS expressed 1.3 times lower than that of DCCP- incorporated cookies with the same substitution ratios. The RIS of DCCP10 was nearly close to the recommended ratio by Dietetic Associations of 3:1. SDF-fortified cookies can bring significant benefits to human health, such as having a lower glycemic index (GI), increasing the retention time of nutrients in digestive tracts, and acting as prebiotics for the growth of probiotics [37].

#### 3.3.3. The Influence of the Partial Replacement of Wheat Flour with DCCP and HDCCP on the Color and Physical Properties of Cookies

The influence of incorporating DCCP or HDCCP at different ratios on the physical appearance and color of cookies is illustrated in Table 6.

The variation in physical properties and color of DCCP or HDCCP-incorporated cookies is demonstrated in Table 6. The substitution of 40% HDCCP cut down the diameter and thickness of cookies by nearly 1.1 and 1.3 times, respectively, compared to DCCP0. The reduction in the diameter of cookies can be ascribed to the WHC of HDCCP being higher than that of wheat flour (1.02 ± 0.05 g water/g db, data not shown), competing for water with the protein and starch of wheat flour in dough preparation. As a result, the dough viscosity increases, limiting the spread of cookies during the baking process [38]. The decrease in thickness can be explained as the diluted gluten content of cookie dough having insufficient durability to resist the air cells’ expansion; then, the gluten network is ruptured in the early stage of the baking process [39]. The thickness of DCCP-incorporated cookies was higher than that of HDCCP-incorporated cookies, probably due to the higher SDF contents of HDCCP. SDF can form hydrogen bonds with water, which prevents wheat flour protein from accessing water to develop gluten networks [40]. Additionally, the higher OHC of HDCCP absorbs lipid in the cookie formula, decreasing the amount of air cell formation during kneading. This phenomenon can partially decrease the vertical expansion of doughs during baking [41].

The addition of 40% HDCCP increased the hardness of cookies by over 1.2 times compared to DCCP0. It can be ascribed to the crystal structure of cellulose from HDCCP inserted into starch during dough preparation and baking [24]. Cookies with the same replacement ratios of HDCCP had a lower hardness than DCCP-incorporated cookies, possibly owing to the IDF content being lowered after cellulase treatment. IDF restricts the enlargement of the gluten dough network framework during the baking process; thus, the hardness of DCCP-incorporated cookies is higher than that of HDCCP-incorporated cookies at the same substitution level [38].

The incorporation of HDCCP or DCCP lowered the L^*^ and b^*^ values while growing the a^*^ value of cookies. The Maillard reaction between the amine groups of protein and the carbonyl groups of carbohydrates raised the darkness and redness of cookies after baking [7]. At the identical replacement ratio, DCCP-incorporated cookies showed a higher L^*^ and a lower ΔE value than HDCCP-incorporated cookies. The darker surface of cookies can be attributed to facilitating Maillard reactions, which result from an increment in the amount of reduced sugar after cellulase treatment [7].

#### 3.3.4. The Influence of the Partial Replacement of Wheat Flour on the Sensory Evaluation of Cookies

Figure 6 expresses the influence of HDCCP or DCCP incorporation on the overall acceptability of cookies.

The overall acceptability of cookies decreased with the increasing amount of incorporated DCCP or HDCCP. These values can have a relationship with high hardness and the darkness of the cookie’s appearance [42].

## 4. Conclusions

The enzymatic hydrolysis process was demonstrated to reduce the RIS of DCCP, owing to a decrease in IDF content and an increase in SDF content. The optimal conditions for the cellulase treatment were discovered at 10.3 g buffer/g of materials, 3.7 U/g of the materials, and 60 min of retention time by using the CCD model. The high values of determination coefficients (R^2^ > 0.97) confirmed the fit of predicted models, and the differences between the forecast and experimental values were not substantial. Cookies made with HDCCP expressed lower RIS, SDF content, and changes in physical properties (color, diameter, thickness, spread factor, and hardness) compared to those with DCCP. Based on the result of sensory evaluation, a 10% HDCCP replacement ratio for fiber-enriched products was the most suitable as it was equal to the conventional product. This study provides a green method for reducing RIS through partially converting IDF into SDF and valorizing the by-product into fiber-enriched cookies.

## Figures and Tables

**Figure 1 foods-11-02709-f001:**
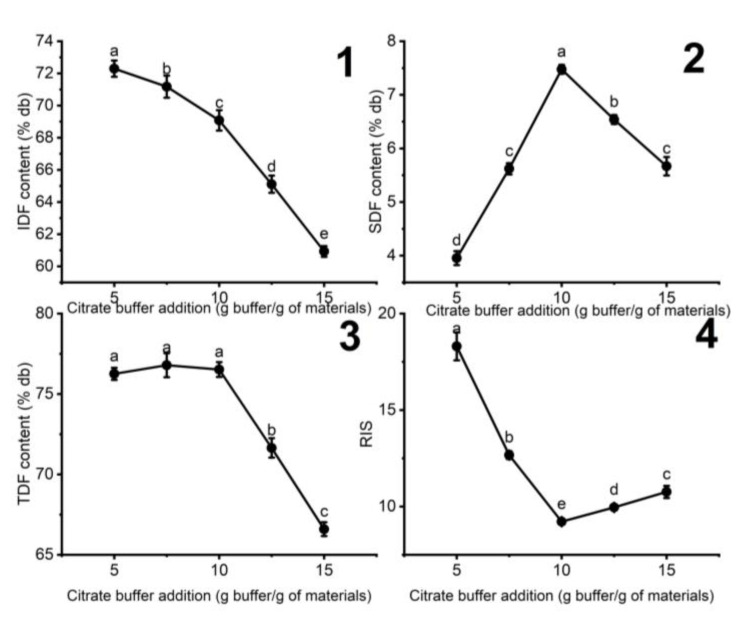
The effect of citrate buffer addition on the (**1**) IDF, (**2**) SDF, (**3**) TDF contents, and (**4**) RIS in HDCCP (enzyme concentration: 5 U/g of the materials, pH 6, 60 min). Different letters indicate significant differences (*p* < 0.05) among the amount of added buffer solution.

**Figure 2 foods-11-02709-f002:**
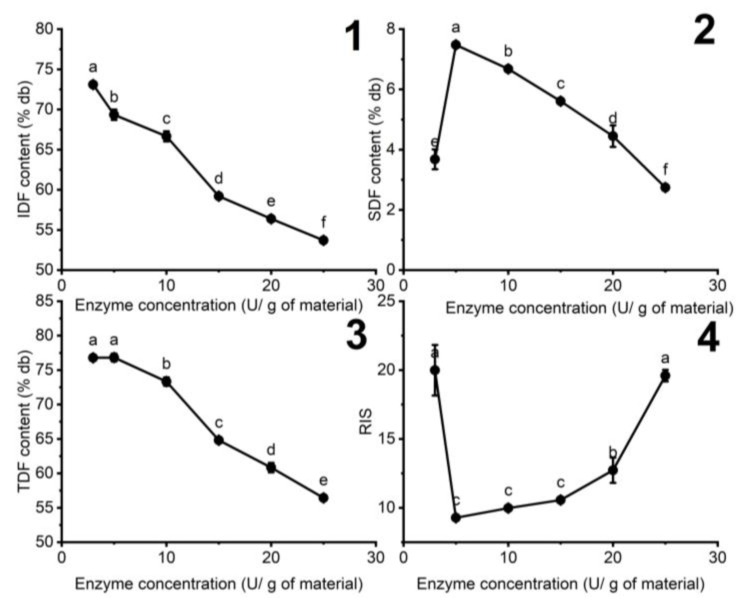
The effect of enzyme concentration on the (**1**) IDF, (**2**) SDF, (**3**) TDF contents, and (**4**) RIS in HDCCP (the added citrate buffer amount: 10 g buffer/g of materials, pH 6, 60 min). Different letters indicate significant differences (*p* < 0.05) among the enzyme concentrations.

**Figure 3 foods-11-02709-f003:**
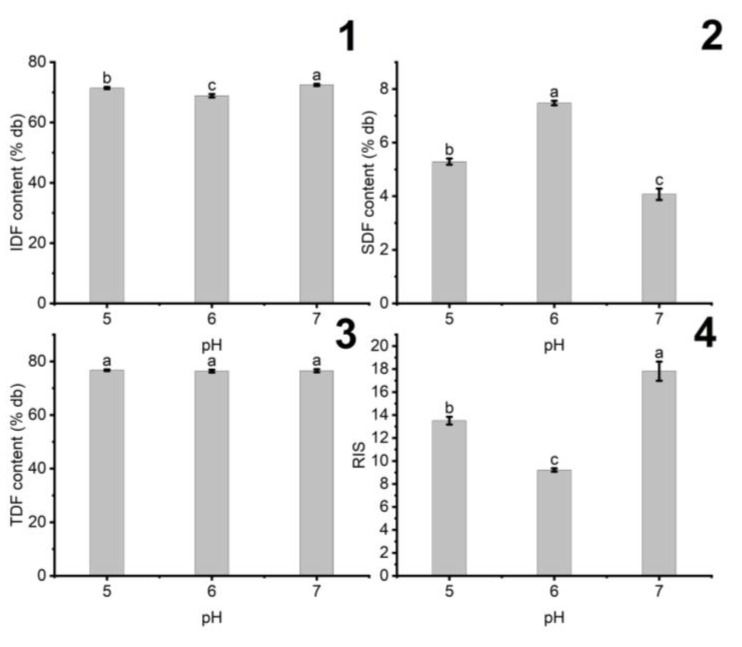
The effect of pH value on the (**1**) IDF, (**2**) SDF, (**3**) TDF contents, and (**4**) RIS in HDCCP (the added citrate buffer amount: 10 g buffer/g of materials, enzyme concentration: 5 U/g of material, 60 min). Different letters indicate significant difference (*p* < 0.05) among the pH values.

**Figure 4 foods-11-02709-f004:**
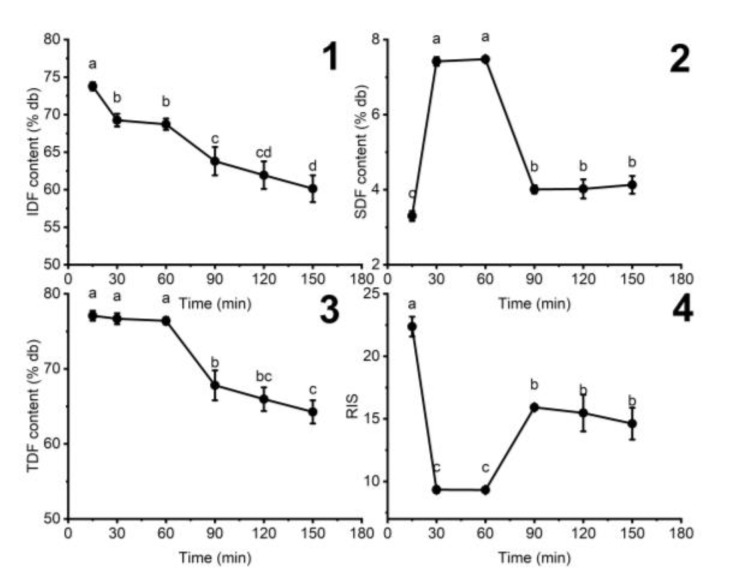
The effect of hydrolyzing time on the (**1**) IDF, (**2**) SDF, (**3**) TDF contents, and (**4**) RIS in HDCCP (the added citrate buffer amount: 10 g buffer/g of materials, pH 6). Different letters indicate significant differences (*p* < 0.05) among the enzyme concentrations.

**Figure 5 foods-11-02709-f005:**
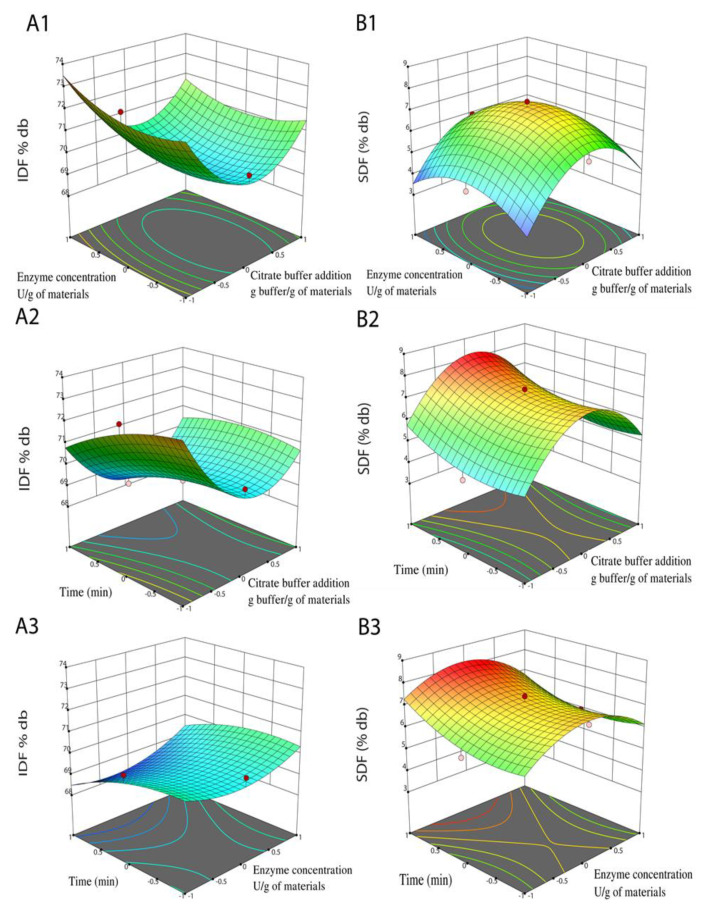
Three-dimensional response surface graphics depict the influence of independent factors on IDF (**A1**–**A3**) and SDF (**B1**–**B3**) contents in the cellulase hydrolysis process. (**A1**–**A3**): The reciprocal relation of citrate buffer addition, enzyme concentration, and time on the IDF content. (**B1**–**B3**): The reciprocal relation of citrate buffer addition, enzyme concentration, and time on the SDF content.

**Figure 6 foods-11-02709-f006:**
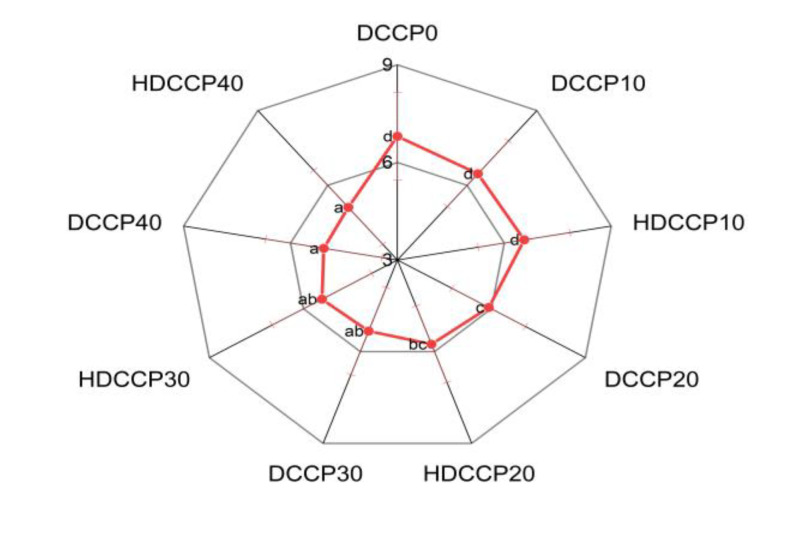
Sensory scores of cookie samples, and the cookie samples with different replacement ratios of DCCP or HDCCP; (a), (b), (c), and (d) denote significantly statistical differences.

**Table 1 foods-11-02709-t001:** The design of experiments for factors and their levels.

Variables	Units	Responses		
		Low (−1)	Middle (0)	High (+1)
X_1_: added citrate buffer	g buffer/g of materials	7.5	10	12.5
X_2_: enzyme concentration	U/g of materials	3	5	10
X_3_: time	min	15	30	60

**Table 2 foods-11-02709-t002:** CCD design and the results of experiments for IDF and SDF contents.

	Independent Factors			IDF%db			SDF%db
Run	X_1_	X_2_	X_3_	Forecast Values	Experimental Values	Forecast Values	Experimental Values
1	−1	−1	−1	73.4	73.1 ± 1.3	3.48	3.52 ± 0.12
2	0	0	−1	69.5	69.9 ± 1.2	7.26	7.09 ± 0.09
3	−1	1	−1	74.0	73.9 ± 1.5	3.76	3.85 ± 0.07
4	1	−1	−1	71.4	71.3 ± 1.5	4.11	4.20 ± 0.06
5	1	1	−1	71.3	71.3 ± 1.5	4.39	4.03 ± 0.05
6	0	0	0	69.3	69.3 ± 1.8	7.01	7.42 ± 0.17
7	0	0	0	69.3	69.3 ± 1.8	7.01	7.42 ± 0.17
8	0	0	0	69.3	69.3 ± 1.8	7.01	7.42 ± 0.17
9	0	1	0	70.3	70.1 ± 1.3	5.84	5.94 ± 0.14
10	−1	0	0	72.3	72.8 ± 1.3	4.53	4.32 ± 0.04
11	1	0	0	71.0	70.6 ± 1.2	5.15	5.24 ± 0.08
12	0	−1	0	69.7	70.1 ± 1.1	5.88	5.67 ± 0.03
13	−1	−1	1	70.8	70.7 ± 1.4	4.75	4.92 ± 0.11
14	−1	1	1	72.1	72.2 ± 1.2	4.37	4.44 ± 0.06
15	1	−1	1	70.8	70.9 ± 1.5	5.37	5.44 ± 0.13
16	1	1	1	71.4	71.7 ± 1.4	5.00	4.81 ± 0.07
17	0	0	1	68.3	68.0 ± 1.7	8.19	8.03 ± 0.22

**Table 3 foods-11-02709-t003:** The quadratic models of independent factors calculated by CCD and their regression coefficients.

	Coefficient	IDF	*p*-Value of IDF	SDF	*p*-Value of SDF
Intercept	Β_0_	69.34	<0.0001	7.17	<0.0001
Linear	B_1_	–0.68	0.0007	0.27	0.0244
	B_2_	0.31	0.0342	–0.07	0.4908
	B_3_	–0.62	0.0012	0.49	0.0012
Interaction	B_12_	–0.18	0.2106	–0.08	0.4676
	B_13_	0.50	0.0069	0.00	0.9770
	B_23_	0.18	0.2143	–0.16	0.1737
Quadratic	B_11_	2.31	<0.0001	–2.21	<0.0001
	B_22_	0.67	0.022	–1.18	0.0003
	B_33_	–0.42	0.1082	0.57	0.0159
DF		9		9	
F-values		29.27		44.30	
*p*-values		<0.0001		<0.0001	
R^2^		0.9747		0.9819	
R^2^_adjusted_		0.9422		0.9587	

Notes: not statistically significant differences (*p* > 0.05); statistically significant differences (*p* < 0.05); highly statistically significant differences (*p* < 0.01); df: degree of freedom.

**Table 4 foods-11-02709-t004:** Experimental and forecast values of IDF and SDF at optimal conditions.

The Independent Factors of the Hydrolysis Process							
An Amount of Added Citrate Buffer g Buffer/g of Materials	Enzyme Concentration U/g of the Materials	Retention Time Min	Dependent Responses	Forecast Values	Experimental Values	Prediction Error%	Rpredicted
10.3	3.7	60	IDF%db	68.21	69.17 ± 0.89	1.39	0.7966
			SDF%db	8.18	7.72 ± 0.47	5.62	0.8963

**Table 5 foods-11-02709-t005:** Chemical composition of cookies.

Samples	IDF%db	SDF%db	TDF%db	RIS	Moisture%	Protein%db	Lipid%db	Ash%db
DCCP0	2.3 ± 0.1 a	1.0 ± 0.2 a	3.3 ± 0.1 a	2.3 ± 0.4 a	3.4 ± 0.1 a	8.4 ± 0.0 d	19.9 ± 0.1 b	1.41 ± 0.01 a
DCCP10	5.0 ± 0.2 bA	1.2 ± 0.1 bA	6.2 ± 0.1 bA	4.3 ± 0.3 bA	3.0 ± 0.3 aA	8.4 ± 0.1 cA	20.0 ± 0.8 bA	1.52 ± 0.04 bA
DCCP20	8.6 ± 0.1 cA	1.3 ± 0.1 cA	9.9 ± 0.2 cA	6.4 ± 0.4 cA	3.0 ± 0.3 aA	8.2 ± 0.0 cB	20.4 ± 0.6 bA	1.59 ± 0.04 cA
DCCP30	14.8 ± 0.2 dA	1.7 ± 0.1 dA	16.6 ± 0.2 dA	8.6 ± 0.3 dA	3.1 ± 0.3 aA	8.0 ± 0.0 bA	20.4 ± 0.4 cA	1.82 ± 0.02 dA
DCCP40	18.4 ± 0.6 eA	1.8 ± 0.0 eA	20.2 ± 0.6 eA	10.0 ± 0.4 eA	3.1 ± 0.2 aA	7.7 ± 0.0 aA	20.6 ± 0.4 cA	1.97 ± 0.01 eA
HDCCP10	4.5 ± 0.2 bB	1.4 ± 0.1 bB	5.9 ± 0.1 bA	3.2 ± 0.3 aB	3.0 ± 0.3 aA	8.1 ± 0.1 cB	19.6 ± 0.1 aA	1.68 ± 0.01 bB
HDCCP20	7.3 ± 0.5 cB	1.7 ± 0.0 cB	8.9 ± 0.5 cA	4.3 ± 0.2 bB	3.0 ± 0.1 aA	8.1 ± 0.1 cB	19.7 ± 0.1 aA	2.69 ± 0.14 cB
HDCCP30	12.8 ± 0.7 dB	2.1 ± 0.0 dB	14.9 ± 0.7 dA	6.0 ± 0.4 cB	3.2 ± 0.1 aA	7.9 ± 0.0 bA	20.3 ± 0.1 cA	2.97 ± 0.08 dB

(a), (b), (c), (d), (e) denotes significantly statistical differences within the same group of cookie samples (*p* < 0.05); (A), (B) expressed significantly statistical differences within the identical amounts of DCCP/HDCCP.

**Table 6 foods-11-02709-t006:** Color and physical properties of cookies.

Samples	L	a	b	ΔE	Diameter mm	Thickness mm	Spread Factor	Hardness kg
DCCP0	62.2 ± 0.3 a	4.8 ± 0.2 a	30.8 ± 0.1 a	0.0 ± 0.0 a	34.0 ± 0.0 d	6.9 ± 0.1 d	5.00 ± 0.1 a	3.1 ± 0.3 a
DCCP10	60.1 ± 0.6 bA	5.1 ± 0.1 bA	26.9 ± 0.2 bA	4.5 ± 0.5 bA	34.1 ± 0.1 dA	6.8 ± 0.1 eA	4.7 ± 0.0 eA	3.2 ± 0.1 aA
DCCP20	56.3 ± 0.6 cA	5.4 ± 0.2 cA	24.5 ± 0.6 cA	8.7 ± 0.5 cA	33.2 ± 0.1 cA	6.6 ± 0.0 cA	5.1 ± 0.0 cA	4.0 ± 0.1 bA
DCCP30	55.2 ± 0.4 dA	6.5 ± 0.5 dA	23.1 ± 0.1 dA	10.5 ± 0.4 dA	32.9 ± 0.1 bA	6.3 ± 0.0 bA	5.2 ± 0.0 bA	4.4 ± 0.3 bA
DCCP40	54.2 ± 0.2 eA	7.1 ± 0.2 eA	22.7 ± 0.2 eA	11.6 ± 0.5 eA	32.5 ± 0.1 aA	5.4 ± 0.1 aA	6.1 ± 0.1 dA	4.9 ± 0.3 cA
HDCCP10	57.3 ± 0.3 bB	5.7 ± 0.6 bB	24.7 ± 0.5 bB	7.9 ± 0.6 bA	34.2 ± 0.2 dA	6.1 ± 0.1 eB	5.7 ± 0.0 bB	2.8 ± 0.1 aB
HDCCP20	54.3 ± 0.7 cB	6.8 ± 0.4 cB	23.7 ± 0.4 cA	10.8 ± 0.6 cB	33.5 ± 0.2 bA	5.8 ± 0.1 cB	5.8 ± 0.1 cB	3.1 ± 0.0 bB
HDCCP30	51.9 ± 0.3 dB	7.1 ± 0.1 cB	22.8 ± 0.2 dB	13.3 ± 0.3 dB	32.7 ± 0.1 aA	5.4 ± 0.0 bB	6.1 ± 0.1 dB	3.6 ± 0.2 cB
HDCCP40	47.1 ± 1.1 eB	8.1 ± 0.7 cB	21.6 ± 0.4 eB	18.0 ± 0.9 eB	32.5 ± 0.2 aA	5.2 ± 0.0 aB	6.1 ± 0.1 dA	3.9 ± 0.2 cB

(a), (b), (c), (d), (e) denotes significantly statistical differences within the same group of cookie samples (*p* < 0.05); (A), (B) expressed significantly statistical differences within the identical levels of DCCP/HDCCP.

## Data Availability

Not applicable.

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
