# Peer review of "Optimized Cellulase-Hydrolyzed Deoiled Coconut Cake Powder as Wheat Flour Substitute in Cookies"

_foods, 2022, doi:10.3390/foods11172709_

Round 1
Reviewer 1 Report
Optimizing enzymatic hydrolysis to increase the soluble dietary fiber of de-oiled coconut cake powder for the partial replacement of wheat flour in an enriched-fiber cookie formula
This study attempts to optimize cellulase treatments on deoiled coconut cake powder (DCCP) to increase the soluble fiber content (SDF), thereby reducing the ratio of insoluble/soluble dietary fiber (RIS). Then the optimally hydrolyzed DCCP (HDCCP) and the DCCP was incorporated at different percentage in baking cookies, textural properties and dimensions of the cookies were then assessed. The cellulase-hydrolyzed coconut cake powder has been found to have higher amount of SDF and lower RIS. The incorporation of HDCCP at 10% in cookies can increase the fiber content without increasing the hardness drastically. There are few scientific and technical issues required the authors’ attentions:
11. Title:
The title may be shortened without affecting the main focus of the study. For example, Optimized cellulase-hydrolyzed deoiled coconut cake powder as wheat flour substitute in cookies
22. Abstract:
o Line 21: Do you mean have 2-fold reduced RIS....?
o Line 22: Do you mean the potential of cellulase-treated deoiled coconut cake powder as a source of soluble fiber to produce .......?
33. Introduction:
o Page 1, Line 39: Please check the grammar.
o Page 1 Line 42: Please mention the meaning of RIS since it was first time mentioned in the text.It is better to provide a list of abbreviation at the beginning, after the abstract.
o Page 2, Line 46: Is manno-oligosaccharide a type of SDF?
o Page 2, Line 48: It is better to replace carbohydrase enzyme with carbohydrate-degrading enzyme (carbohydrase indicates it is an enzyme), no need to add an enzyme behind the 'carbohydrase'
44. Material and Methods:
o Page 2 Line 17: Do you mean on the dried weight basis? or The composition is based on the dried sample?
o Page 2 Line 82 to 104: 2.2.1 Hydrolyzing DCCP with Cellulast 1.5L No description on how the enzyme-induced hydrolysis was done? But The description was on how to incorporate/substitute wheat flour with DCCP or HDCCP to form cookies.
o Page 2 Line 87: B equivalent to 0% replacement? same as the description in Line 89
o Page 3 Line 108 to 110: Should you describe how do you choose the range of the added water, enzyme concentration, time of reaction? How did you stop the enzymatic hydrolysis? Instead of using added water, why did not you use the solid to liquid ratio because no description on the 7.5, 10 and 12.5 g, was it based on 1 g of DCCP? It would be better you include it in the introduction the rational of adding low and high amount of water as a studied factor in optimization of cellulase action on DCCP.
o Page 3 Line 109: Should you include the description of any software/programme that you used?
o Page 3 Line 126: You need to include ash content determination in order to quantify SDF using enzymatic-gravimetric method.
o Page 3 Line 128: Do you mean you assay for Cellulast 1.5L activity? Is this part of the chemical analysis? Reference 18 and 19 seems not relevant.
o Page 3 line 136: How did you measure the spread factor as part of the result in Table 6?
55. Results and Discussion
Page 3 Line 148: 3.1 The Influence of water……(Result)..No description on the level of cellulase used and the time of hydrolysis under Materials and Methods 2.2?
· Page 3 Line 154: Is this RIS (9.22) acceptable to make the HDCCP a perfect wheat substitute?
· Page 3 Line 155: typo error. Waster à water
· Page 3 Line 158: Does DCCP contain high level of lignin?
· Page 5 Figure 2: The title of the figure should be rephrased to prevent repeated terms. For example, (1) IDF; (2) SDF (3) TDF and (4) RIS content of hydrolyzed DCCP as a function of enzyme concentration (added water: 10g, pH 6, 60 min). Different letter denote...... (n =?)
· Page 5 Line 178: Further elaboration is required. if the IDF also was degraded by cellulase action, how come the RIS increased with enzyme concentration?
· Page 5 Line186: Please cite a reference. State what are Asp 252, Asp 392 (from the DCCP?)
· Page 5 Line 193: If this is true, should the TDF also decrease?
· Page 7 Line 209: The text has to be put before the figure!
· Page 9 Figure 5: Would it be better you arrange the figures side by side A1 beside B1, A2 beside B2 and A3 beside B3?
· Page 10 Line 278: Do you wish to cite a reference here?
· Page 10 Line 284: is the WHC of DCCP significantly higher as compared to the HDCCP ?
· Page 11 Line 315: Do you mean lower WHC of HDCCP increases the dough viscosity?
· Page 11 Line 326: Would 50% DCCP addition also increases the hardness of cookies by 1.2 fold?
· Page 12 Figure 6: Results of sensory test but no description on sensory test under Method and Material.
76. Conclusions
Some vocabulary such as growth in SDF content may be modified for clarity.
7. Reference listing:
Inconsistent and some are not following the format: For example, the title of a journal article should only capitalize the first letter of the first word, right?
Some references seem not relevant to the statement that cited them. For example, reference 18 and 19.
Language:

Author Response
Dear Reviewers
Thank you for your detailed recommendation and suggestion for our manuscript, we revised carefully the manuscripts. We also provided detailed explanations for your several comments in the enclosed file
Please see the attachment
Sincerely

Reviewer 2 Report
Manuscript foods-1884838 has an interesting topic that has practical relevance, as well. The manuscript is generally well structured. Introduction section is superficial, but highlights well the relevance of the study. Applied physical tests (TPA, colorimetry) and analytical methods are adequate and described well in the manuscript. But details of enzyme treatment are missing. The manuscript contains interesting results, but need revision before publishing.
Comments, suggestions:
Please give the acronyms in the first place in the manuscript (see line 42, RIS).
Please discuss the applicability of Celluclast, optimum process parameter range and dosage in more details in the Introduction section, as well.
Why the authors did not test the applicability of other enzymes/enzyme blends?
Please give detailed information related to enzyme, and enzyme dosage, mixing etc. in the methodology section.
Details of sensory analysis are missing.
Please give the percentage of added water instead of amount (because it is depended on the amount of the product, in my opinion).
In table 1 authors give 12.5 g as highest level for variable x1. But Figure 1 show highest dosage. Please explain it. Figure 2 (show the enzyme dosage effect, see Table 1-Fig2, as well).
Line 193-194 need reference(s) (’.. probably due to..’)
Please improve the visibility/quality of figure 5.
Please discuss briefly the price/economy of enzymatic hydrolysis process, as well.
How affect the microbial state/food safety the enzymatic hydrolysis (microbial count, water activity etc).
Author Response

(The authors gave the same response as above.)

Reviewer 3 Report
The manuscript requires thorough revision in language. Scientific design seems vague. Kindly rewrite the manuscript with clear experimental designs its outcomes. It is very difficult to follow. Title. abstract, introduction and results are not in continuum. Looks like initially the design of experiment was something else which later on changed. That makes it very difficult to follows. Clear objective and aim of the experiment should be kept in mind during writing. Lot of experiment does not necessarily mean that the work is good. Experimentation must be done formulated as per the requirement of the problem/hypothesis. The basic concept or hypothesis is interesting and lot of efforts have been done to accomplish it but scientific writing is lacking which is making the work very difficult to understand. Kindly put in equal efforts in formulating the manuscript as much as done for experimentation.
Author guidelines must be revisited to correct the manuscript

Author Response
Dear Reviewers
Thank you for your recommendation and suggestion for our manuscript, we revised carefully the manuscripts. We also revised the language and style of my manuscripts detailedly
Please see the attachment
Sincerely

Round 2
Reviewer 1 Report
1. Page 3: Please explain how did you obtain HDCCP through sieving the DCCP through a 40 mesh.
2. Page 5: Textural attributes of the dough such as hardness, cohesiveness and springiness were measured but no description and discussion on the observations/results. In addition, no description on the type of test mode such as compression or tensile, etc as well as the probe moving speed was stated.
3. Please check the grammar as commented in the attached document.

Author Response
Dear Reviewer
Thank you for your suggestions and comments that help us improve the manuscript. Several featured comments were responded in the enclosed file
Please see the attachment
Sincerely

Reviewer 2 Report
The authors have revised the manuscript thoroughly accoding to the reviewers' comments and suggestions and provided etailed answers for the reviewers questions. Repharisng, amendments with more detailed information and discussion made the manuscript more complete and clear. The overall scientific quality of the mnauscript has improved significantly due to the revision. I agree and accept all modifications made by the authors.
Author Response
Dear Reviewer
Thank you for your suggestions and comments that help us improve the manuscript, and accept the modifications
Please see the attachment
Sincerely

Reviewer 3 Report
The manuscript has been thoroughly revised and improved according to the reviewers suggestion.
Author Response

(The authors gave the same response as above.)
